# Study on Trans-Boundary Water Quality and Quantity Ecological Compensation Standard: A Case of the Bahao Bridge Section in Yongding River, China

Yizhuo Wang [1], Rongjin Yang [2], Xiuhong Li [1,*], Le Zhang [2], Weiguo Liu [3], Yi Zhang [1], Yunzhi Liu [1] and Qiang Liu [1]

[1] State Key Laboratory of Remote Sensing Science, College of Global Change and Earth System Science, Beijing Normal University, Beijing 100875, China; wyz999@mail.bnu.edu.cn (Y.W.); zhangyi_hugo@mail.bnu.edu.cn (Y.Z.); 202021490017@mail.bnu.edu.cn (Y.L.); toliuqiang@bnu.edu.cn (Q.L.)

[2] Chinese Research Academy of Environmental Sciences, No.8, Da Yang Fang, An Wai, Chaoyang District, Beijing 100012, China; yangrj@craes.org.cn (R.Y.); zhangle@craes.org.cn (L.Z.)

[3] Urban Drainage Maintenance Management of Linyi City in Shandong Province, No.86, Liuqing Dong Road, Lan Shan District, Linyi 276000, China; liuweiguo197707@163.com

[*] Correspondence: lixh@bnu.edu.cn; Tel.: +86-136-2116-6693

**Abstract:** Watershed ecological compensation, as an important means to protect the environment and promote the sustainable and coordinated development of upstream and downstream has wide concern in China. At present, the compensation accounting method only assesses water quality. When applied to some northern rivers represented by the Yongding River, which are facing water shortage, the assessment of water quality indicators alone cannot effectively compensate the ecosystem service providers for their expenditure on the environment. This paper proposes a transboundary water quality and quantity ecological compensation standard model, which couples the water quality ecological compensation standard of pollutant reduction and the water quantity ecological compensation standard based on the restoration cost method. We set up two scenarios using the model to calculate the amount of compensation payable under the actual scenario in 2018, which is USD 68.2 million. The amount of compensation under the local environmental policy target scenario is USD 10.6–82.668–529 million. It was concluded that the funds obtained from this model can cover the rehabilitation cost and meet the benefits of the upstream and downstream, making compensation funds more reasonable. However, based on the cross-sectional assessment, there is still a lack of integrity and comprehensiveness for the river basin. The development of watershed ecological compensation should move from the game of upstream and downstream interests to a win–win situation.

**Keywords:** ecological compensation; ecological compensation standard; payment for ecosystem services (PES); Yongding River

## 1. Introduction

Due to the mobility of water and the integrity of the watershed, different areas within the watershed are interrelated in the development, utilization, pollution management, treatment, and protection of water resources. However, in the process of transforming resources into productivity and economic benefits, they are different from each other due to the imbalance of regional developments. Through the PES mechanism, wide concern has been given to quantitatively alleviate the current water environment pollution, unreasonable development and utilization of water resources in the watershed, to coordinate the environment of different regions in the watershed with the sustainable development of economy and society, and to protect the ecosystem safety in the watershed. With the increasing international attention toward the ecological value of river watersheds, some

pilot projects of river watersheds ecological compensation have been carried out in developing countries with the support of international organizations such as the International Fund for Agricultural Development (IFAD), The Cornell International Institute for Food, Agriculture, and Development (CIIFAD). The Pago por Servicios Ambientales (PSA project) in Costa Rica, Central, and South America, was the first project to explore the payment for environmental services of the river watershed ecosystem, and was successfully operated for a long time [1]. The Working for Water Program located in South Africa, Africa, has become a model for coordinating fairness, efficiency and sustainability in similar projects in African countries by giving economic value to ecosystem services and eliminating invasive plants [2]; The watershed ecological compensation projects in Sum Berjaya, Brantas, and Kapuas Hulu regions in Indonesia, Asia, have established a set of efficient trans-boundary watershed ecological service consultations by ensuring the enlightenment of residents in the watershed and the participation of stakeholders in the decision-making negotiation mechanism [3]. Similar special organizations and projects responsible for the payment of ecological services have gradually grown and solved some practical problems of ecological compensation in transboundary watersheds. For example, the Rhine is a famous cross-country river that flows through Switzerland, France, Germany, Luxembourg, and the Netherlands. The countries established the Watershed Management Commission (ICPR) and signed the Rhine Protection Convention and the Rhine Action Plan, which used biological indicators (salmon return to rivers) to test river water quality. They emphasized the polluter pays principle and paid attention to the role of water price on water quality protection and water saving. The Elbe River runs through two countries, upstream in the Czech Republic and downstream in Germany. After 1990, Germany and the Czech Republic reached an agreement to jointly control the water pollution of the Elbe River and set up a bilateral cooperative organization. Using the sewage fees paid by the citizens, the financial loans, research grants, and compensation were paid by the downstream Germans to the upstream Czechs so that the pollution problems were jointly solved. The Columbia River, a transboundary river located in North America, has also carried out PES projects related to the development and utilization of water resources. It was entrusted by the governments of upstream Canada and the downstream the United States to the international committee to carry out the technical investigation and put forward solutions. The International Commission took into consideration various factors such as water resources, hydropower resources, flood control, and disaster mitigation, and proposed a win–win solution based on benefit sharing and benefit compensation through quantitative calculation of project investment, flood control benefits, and power generation benefits. PES is widely called ecological compensation in China; both are consistent. With the in-depth understanding of PES theory and mechanisms, various parts of China have successively carried out a series of ecological compensations aimed at water pollution control, ecological water transfer, and water quality and quantity based on the measurement and calculation of regional pollutant discharge. The current studies and programs focus on water pollution ecological compensation. Successful implementation of the cases includes: the Xin'an River Watershed in Zhejiang-Anhui implements a tripartite compensation agreement between the central government and the local governments of the two provinces. Using the pollutant indicators of the river watershed to construct an ecological compensation index to form the "Xin'an River Model", the water quality of transboundary sections is assessed, and the scale of compensation funds is determined according to the size of the index [4]. The upstream and downstream "bidirectional watershed ecological compensation" was carried out in Tingjiang River Watershed of Fujian Province, and the direction and amount of compensation were determined according to the cross-section assessment of whether the pollutant concentration exceeded the national standard [5]. Meanwhile, scholars have studied the ecological compensation standard of water pollution: Lu et al. built an econometric model to calculate the amount of ecological compensation for water pollution in the watershed [6]. Liu et al. used the one-dimensional water quality model and the pollution loss function method to calculate the pollution loss in each water function area of the Xiangjiang River

and quantify the scale of ecological compensation funds based on the pollution loss cost [7]. Cheng et al. provided a comprehensive evaluation framework for the economic evaluation of watershed sewage treatment. The total pollutant control model based on the opportunity cost approach was used to compensate for the water quality and the information entropy method was used to allocate the funds [8]. Water transfer ecological compensation is mainly aimed at river watersheds where water resources are unevenly distributed and currently focuses on the study of water transfer ecological compensation standards. Dong et al. used the direct and opportunity cost of ecological protection and environmental protection in Shiyan City, the main source of the South-to-North Water Diversion Project, minus state-funded ecosystem service payments and internal effects to determine the scope and standard methods, and proposed internal effects to define the water source zone [9]. Zhang et al. used the ecosystem service value evaluation method to calculate and analyze the ecological compensation standard of the water source area and the proportion of compensation funds among provinces in the water area involved in the middle line project of the South-to-North Water Transfer Project [10]. Based on scientific water distribution by the AHP and entropy methods, Geng et al. calculated the comprehensive compensation amount of water in each region of the watershed according to the way of compensation step by step by allocating the upstream protection cost according to the amount of water that should be allocated, adjusting the actual water use and water-saving contribution of each region according to the way of compensation step by step [11]. In addition to the standard model established for a single evaluation index of ecological compensation for water quality or water quantity, some scholars proposed a standard model for ecological compensation for water quality and water quantity based on the measurement of regional pollutant discharge for areas with poor water quality and water quantity shortage: Xu et al. proposed a method for calculating the amount of ecological compensation across administrative boundaries based on river water quality and volume, and adopted the comprehensive pollution index method to calculate the amount of water pollution compensation across boundary sections [12]. Wang et al. assessed the flux of COD pollutants and calculated the ecological compensation of water pollution among counties (cities) in the Qiantang River Watershed [13]. Tang et al. established the Taihu Lake water pollution compensation model from the point of view of total pollutant control and calculated the total pollutant intake assessment and the lake water quality assessment respectively [14]. Liu et al. constructed the interprovincial ecological compensation model based on pollutant quantity and grey footprint method and calculated the ecological compensation standard of 11 provinces in the Yangtze River Economic Belt [15]. Although the current water quality and quantity ecological compensation pilots take into account the two factors of water quality and water quantity, there is no basis for setting the compensation amount. The standard model of ecological compensation for water quality and quantity studied by previous researchers is limited to flux assessment. Although it solves the problem of a single target and one-sided assessment based on the annual average pollutant concentration of transboundary sections, it is mainly restricted to water resources. Concerning rivers, especially in the northern arid and semi-arid regions, the weight of water quality in the goals of ecological compensation in the watershed should be considered comprehensively based on actual conditions. Given the obvious time lag effect of ecological environmental protection and construction, many tasks cannot be effectively displayed on the annual time scale. The assessment of annual cross-sectional indicators can easily lead to short-sightedness and biased decisions. Therefore, we summarized the current cases of watershed ecological compensation implemented in China and the deficiencies of previous studies on the compensation mechanism. A novel and operable ecological compensation calculation model for transboundary watershed has been provided in this paper. Taking the Yongding River watershed in northern China as the study area, based on the original ecological compensation accounting method, which mainly focused on basin water quality evaluation, a standard ecological compensation model combining water quality and quantity was established by introducing water resource evaluation index. Based on the actual situation of water environment and water

resources in the Yongding River Basin, the relationship between the above two indexes has been discussed to explore the calculation method of ecological compensation for different regions. In addition to the calculation of ecological compensation, this paper also made a preliminary study on the stability of water quality and water quantity in water-deficient areas. It highlights the necessity of the water quality and water quantity compensation and provides a new way for decision-makers to negotiate with stakeholders.

## 2. Materials and Methods

### 2.1. Study Area

Yongding River is the mother river of Beijing, the capital of China. As the largest part of the Haihe River system in North China, the river is about 747 km long, with a vast drainage area of 47,016 m$^2$ [16]. There are many towns and densely populated cities along the route. The upstream flows through China's water-deficient provinces—Inner Mongolia, Shanxi, and Hebei—and the downstream flows through the Chinese capital Beijing and the municipality of Tianjin to join the Bohai Sea. The Bahao Bridge section (Figure 1) is the first state-controlled section after the intersection of the two major tributaries of Yongding River (Yang River and Sanggan River). It is a transboundary section between the upstream Hebei Province and the downstream Beijing. Additionally, it is the entrance section of Guanting Reservoir, an important source of drinking water in Beijing. With the rapid development of the economy and society, the water consumption of industry, agriculture, and urban residents in Shanxi, Inner Mongolia, and Hebei in the upper reaches of the basin increased rapidly. The water inflow from the upper reaches of Guanting Reservoir decreased rapidly due to the construction of reservoirs. At the same time, affected by the discharge of wastewater from coastal power plants, coal mines, and farmland irrigation, the water quality of the watershed has deteriorated, and the eutrophication trend is obvious [17]. As an important node of the Yongding River watershed in the Beijing–Tianjin–Hebei region, it is difficult to guarantee the water quality and quantity of the Guanting Reservoir, which has a serious impact on product development, water safety, and the maintenance of ecological service function in the downstream region. Therefore, the water quality and hydrological data of the Bahao Bridge section of the Guanting Reservoir are an important basis for reflecting the change of transboundary water quality of the Hebei–Beijing section of Yongding River, and can provide an important reference for the follow-up "ecological compensation of Yongding River".

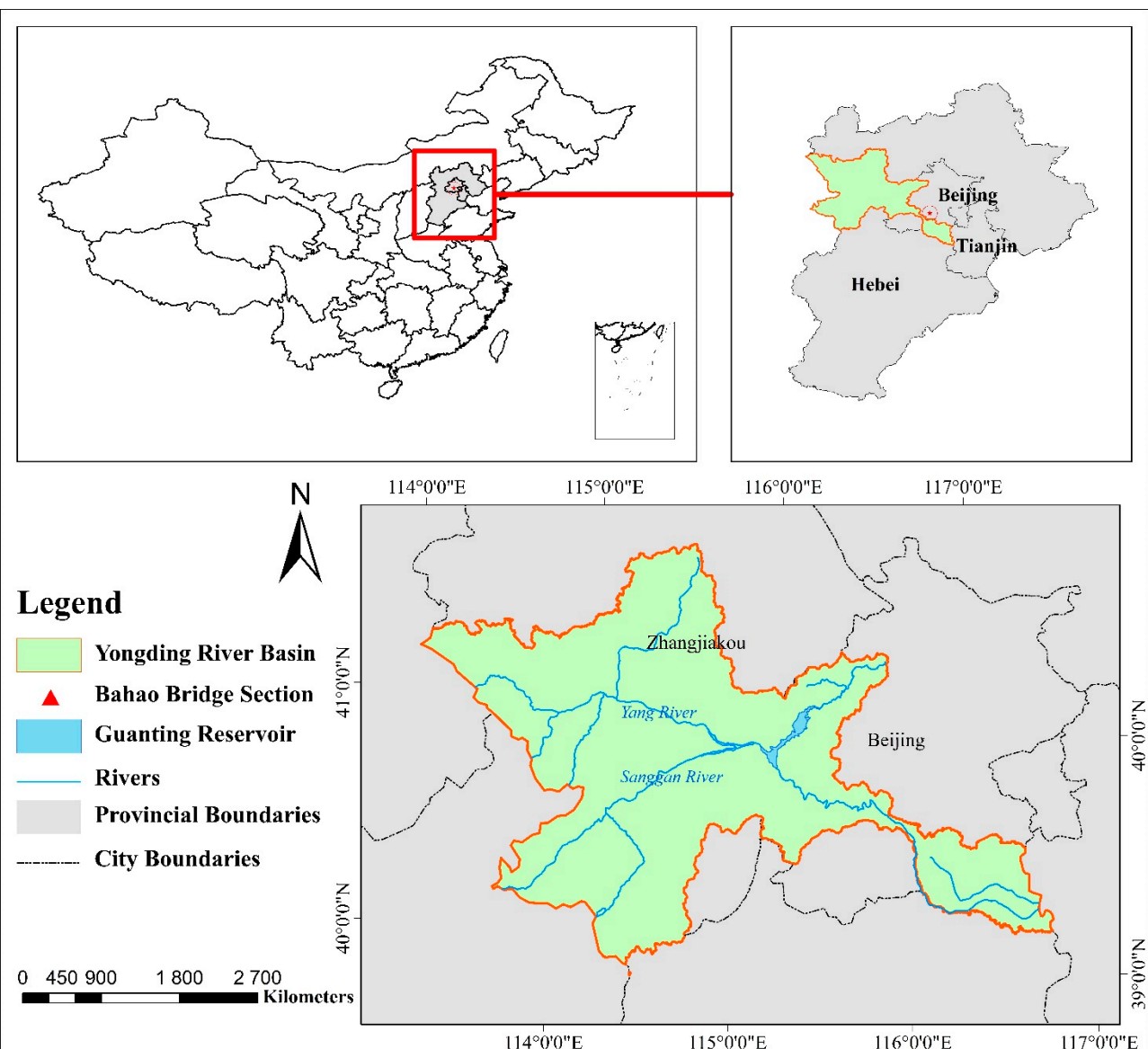

**Figure 1.** The location of the Bahao Bridge Section in the Yongding River watershed.

*2.2. Conceptual Framework for Joint Ecological Compensation for Trans-Boundary Section Quality*

A new ecological compensation model is established in this study for watersheds with cross-administrative units: the impacts of upstream areas on downstream rivers in water resources and water environment are concentrated on the cross-section of rivers set up in the two administrative regions. The total amount of ecological compensation is calculated by considering the number of water resources and the quality of the water environment (Figure 2).

$$W_t = W_p + K \times W_y \tag{1}$$

where $W_t$ is the total amount of ecological compensation; $W_p$ is the amount of ecological compensation for the pollutants of the watershed; $W_y$ is the amount of ecological compensation for the water yield; $K$ is the adjustment coefficient between water quality and quantity compensation.

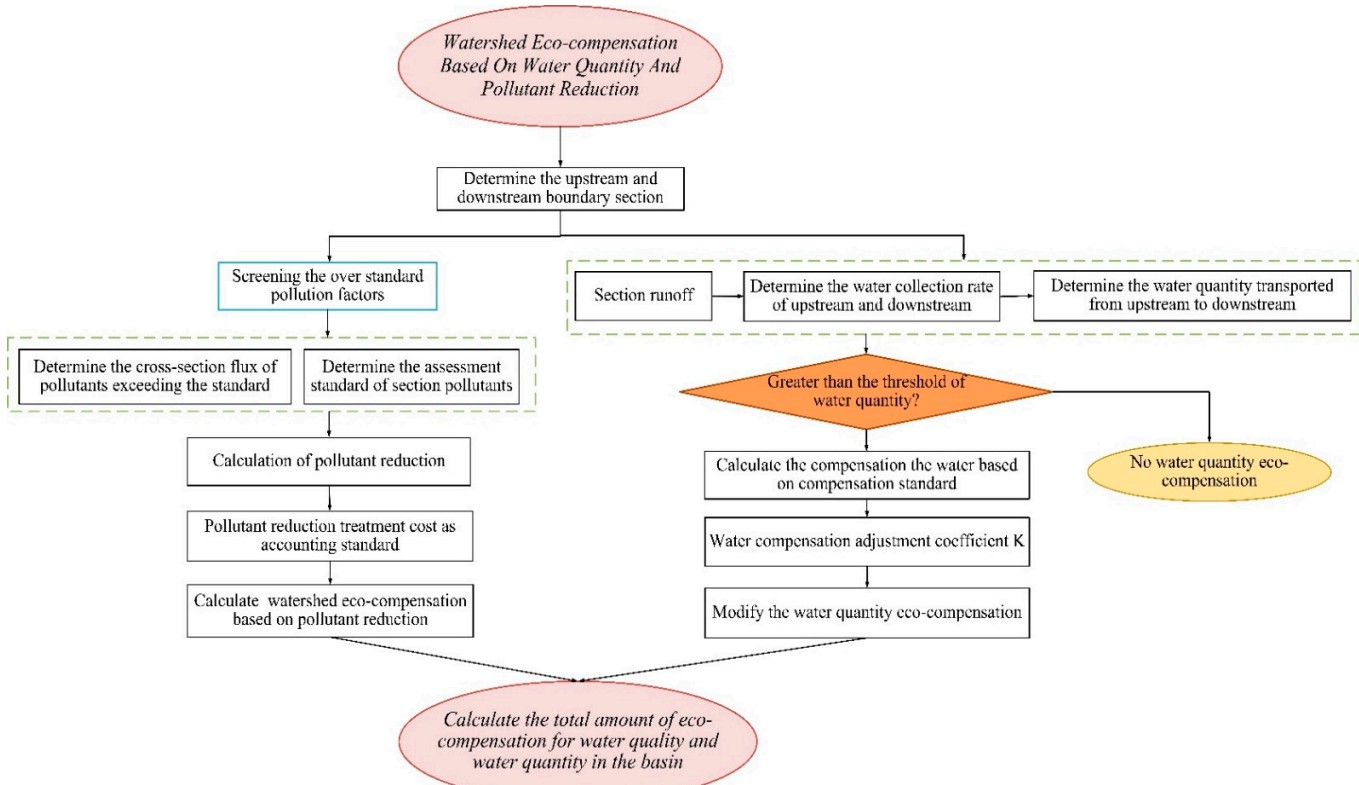

**Figure 2.** A conceptual model of watershed ecological compensation based on transboundary water quality and quantity.

2.2.1. Water Quality Ecological Compensation Standard Model Based on Pollutant Reduction

From the perspective of pollution treatment cost, the calculation method of the water quality compensation standard is based on the requirements stipulated in the functional zoning of the surface water in the watershed: (1) First, select the monitoring section of the transboundary area of the river watershed as the assessment section. According to the surface water function area of the assessment section, determine the water quality target and corresponding pollutant water quality standards of the area by the National Standard of the People's Republic of China GB3838-2002 Surface Water Environmental Quality Standard Limit. (2) According to the monthly data of pollutant concentration published by the national control section monitoring station and the water quality target standard limit of the assessment section, the over-standard rate of each pollutant in the recent three years was calculated. The over-standard rate of the three years was 0%, which means that the pollutant is included in the list of pollutants for water quality compensation assessment.

$$r = \frac{N_i}{N_T} \times 100\% \qquad (2)$$

where $r_i$ is the over standard rate of pollutant $i$; $N_T$ the total monitoring times of pollutant $i$; $N_i$ is the times exceeding the standard of pollutant $i$. (3) After determining the excess pollutants, combines with the actual transit water volume, pollutant concentration value, and the pollutant water quality target standard limit of the assessment section. According to Equations (3) and (4), it can be concluded that the actual situation is compared with the area where the watershed is located. Water quality target requirements, the total amount of pollutants reduced by the main body of ecological compensation in the upstream watershed.

$$P_i = P_{i0} - P_{ia} \qquad (3)$$

$$P_{i0} = \sum\nolimits_{n=1}^{12} c_{0i} \times V_n \qquad (4)$$

where $P_i$ is the reduction of excess pollutant $i$, $P_{i0}$ is the total amount of transit of excess pollutant i under the surface water functional zoning water quality target limit; $P_{ia}$ is the total amount of excess pollutant $i$ transiting under actual conditions; $c_{0i}$ is the standard limit concentration of pollutant $i$; $V_n$ is the monthly actual transit water volume of the transboundary section. (4) Since the upstream area has reduced the discharge of pollutants and reduced the pressure on the water environment capacity of the downstream area, the downstream area of the river watershed, as the main body of ecological compensation in the river watershed, should pay corresponding fees to the upstream area as compensation.

The standards of the ecological compensation for the river watershed water quality should be based on the treatment costs of different types of pollutants and environmental capacities, which is also the basis for the formulation of the environmental protection tax [18]. The environmental protection tax collection standards in various parts of China take into account the treatment cost of pollutants and the level of local economic development [19]. It can provide a practical reference for the establishment of a watershed ecological compensation standard based on pollutant reduction. Accordingly, this scheme adopts the equivalent value of all kinds of pollutants stipulated in the Environmental Protection Tax Law of the people's Republic of China [20] as a reference (Table 1) and formulation of the ecological compensation standard of watershed water quality in combination with the amount of pollutant collection tax in the specific watershed. According to the equivalent value table, the excess pollutant reduction amount is converted into the equivalent table. According to the amount of pollutant tax payable in the downstream area of the watershed (in dollar/equivalent), the cost of excess pollutant flux treatment is calculated as the ecological compensation amount of water quality.

$$W_p = \sum_{i=1}^{n} P_i \times EV_i \times C_P \tag{5}$$

where $W_P$ is the amount of water quality ecological compensation based on the reduction of pollutants and the cost of pollutant treatment; $P_i$ is the reduction of excess pollutants; $EV_i$ is the equivalent value of pollutant i specified in the environmental tax law; $C_P$ is the pollution tax in this watershed.

**Table 1.** Pollutant equivalent value in the environmental protection tax law of China.

| Type of Pollutant | Equivalent Value (kg) |
|---|---|
| Suspended Solids (SS) | 4 |
| Biochemical Oxygen Demand (BOD$_5$) | 0.5 |
| Chemical Oxygen Demand (COD) | 1 |
| Ammonia Nitrogen NH$_3$-N) | 0.8 |
| Total Phosphorus (TP) | 0.25 |
| Fluoride (F$^-$) | 0.5 |
| Total Organic Carbon (TOC) | 0.49 |
| Total Mercury (Hg) | 0.0005 |
| Total Chromium | 0.04 |
| Total Cadmium | 0.005 |
| Petroleum | 0.1 |
| Volatile Phenol | 0.08 |
| Anionic Surfactant (LAS) | 0.2 |

2.2.2. Constructing a Standard Model of Water Ecological Compensation Based on Restoration Cost Method

The ecological compensation of the transboundary watershed involves two areas upstream and downstream of the watershed, so the monitoring section near the junction of the two regions should be selected as the monitoring section of the watershed ecological compensation in the process of carrying out the ecological compensation of water quantity. The annual runoff is calculated by using the monitoring section flow data. A flow diachronic curve method is used to calculate the outbound water volume $V_0$ in the upper reaches

of the river in a special dry year (exit rate 95%) as the threshold for water ecological compensation funds. The flow diachronic curve method is based on historical flow data to construct the monthly flow duration curve, and the flow $Q_p$ corresponding to a certain cumulative frequency is regarded as the ecological flow. The frequency $P$ of $Q_p$ is taken as 95%, and $Q_{95}$ is the commonly used low flow index or extremely low flow condition index as the minimum flow to protect the river.

This method first sorts the historical traffic series from large to small, such as $q_i$, $i = 1$, 2, . . . , n.

Where $q_1$, $q_n$ are the maximum and minimum flow values in the sequence.

Then calculate the cumulative frequency according to Equation (6), and draw the flow duration curve. Finally, according to the flow duration curve, the flow corresponding to the 95% cumulative frequency is taken as the threshold $V_0$ of water ecological compensation funds.

$$p_i = p(\text{Q} > q_i) = \frac{i}{n+1} \tag{6}$$

Compensate for the difference between the actual exit water volume $V$ and the water volume compensation threshold $V_0$, the water volume $\Delta V$. At the same time, taking into account the transboundary water replenishment in the watershed, in the process of water replenishment, there are losses such as infiltration and evaporation of water in the river. Therefore, this paper introduces the water recovery rate $\eta$ in the process of estimating the amount of water participating in ecological compensation and uses it as a parameter reflecting the downstream water resource utilization efficiency.

$$W_y = \frac{\Delta V}{\eta} \times C_y \tag{7}$$

$$\eta = \left( \frac{W_L}{W_0} \right)^{\frac{1}{L}} \tag{8}$$

where $W_y$ is the amount of ecological compensation for the water yield; $\Delta V$ is the number of water resources participating in ecological compensation; $\eta$ is the water collection rate in the downstream area; $C_y$ is the compensation standard for transboundary water in the area where the watershed is located. $W_L$ is the downstream water delivery; $W_0$ is the upstream water delivery; $L$ is the river length.

The water compensation standard is the unilateral price (m$^3$/dollar) of the water supplied to the downstream area after the upstream area exceeds the water threshold. The water received downstream comes from the runoff replenishment under the natural state of the river and the man-made water-saving projects implemented in the upstream area. For the northern water-scarce rivers, the amount of water resources consumed by the economic and social development of the upstream cities is much higher than the water utilization limit in the area. Therefore, it is difficult for downstream regions to obtain water resources to meet their own development needs. The restoration cost method is also known as the alternative engineering method. The restoration cost method is to construct a series of projects artificially to replace or restore the original ecological benefits after the ecosystem is damaged. The cost of constructing new projects is used to estimate the economic loss caused by the destruction of the ecosystem [21–24]. Applying to the ecological compensation of water shortage in rivers, to restore the water resources of the river, various water-saving projects are planned to be carried out in the upstream area, and the amount of water saved and the project investment amount after the implementation of the water-saving projects are estimated. Link the cost of the water-saving project to the water compensation standard, and calculate the cost of saving water for one party as the water transfer price in the water compensation standard.

$$C_y = \frac{ESV_{Vm}}{V_m} \tag{9}$$

$$ESV_{Vm} = \sum C_m (m = 1, 2, 3 \cdots n) \tag{10}$$

where $C_y$ is the compensation standard of transboundary water quantity in the area where the watershed is located; $V_m$ is the amount of water yield added after the restoration project; $ESV_{Vm}$ is the ecological benefit value for the increased amount of water resources; $C_m$ is the construction cost of m project in the restoration project.

2.2.3. Water Quality–Water Quantity Compensation Amount Adjustment Coefficient

Integrate the two variables of transit water volume $\Delta V$ and transit pollutant reduction amount $P_i$ to calculate the ecological compensation amount $W_t$ in the watershed, and set the watershed ecological compensation fund accounting intervals for the water quality dimension and the water volume dimension respectively. Within the set accounting interval, different weights of water quality compensation and water quantity compensation can be determined according to the different requirements for water resources and water environment quality in the ecological compensation area of the river basin. So that it can achieve a more accurate needs matching various stakeholders and realize the purpose in the ecological compensation.

Combining the specific natural conditions in the river watershed and the actual needs of all stakeholders in the ecological compensation of the river watershed, the study determines the weighting relationship between water quality and water quantity in the calculation method of ecological compensation in the river watershed. Since different regions have different requirements for the water environment quality of the watershed and the number of water resources in the watershed, this method sets the water compensation amount adjustment coefficient K, and adjusts the amount of water compensation funds according to the different demand for water resources in the watershed, to better meet the watershed the needs of various stakeholders within. This method selects the ratio of water resources per capita in the water supply area to the water receiving area as the K value. When the ratio of per capita water resources in the upstream and downstream areas is greater, it indicates that the downstream areas have a more urgent demand for water resources, and the amount of water compensation should be increased accordingly to encourage upstream areas to increase the supply of water resources.

$$K = \frac{PCWR_u}{PCWR_d} \tag{11}$$

where $PCWR_u$ is the per capita water resources in the upstream water supply area; $PCWR_d$ is the per capita water resources in the downstream water receiving area; K is the adjustment coefficient of the amount of water compensation.

3. Results

This study sets two different scenarios: (a) the amount of ecological compensation is calculated based on the actual transit water volume and actual pollutant transport volume of the Bahao Bridge section of Yongding River in 2018; (b) According to Yongding River watershed, the Ministry of Water Resources, Forestry Bureau, the National Development and Reform Commission jointly issued the "Overall Plan for Comprehensive Treatment and Ecological Restoration of Yongding River" (Overall Plan for short). The study set another scenario that is based on the Overall Plan and the present situation of development in this watershed. According to the hydrological series from 1956 to 2010, the transboundary water volume of Bahao Bridge was calculated under different guarantee rates—normal years (guarantee rate 50%), ordinary dry years (guarantee rate 75%), and special dry years (guarantee rate 95%), and the ecological compensation amount of the watershed were calculated after the water quality was raised to Class II.

### 3.1. Calculation of Water Quality Ecological Compensation Amount Based on Pollutant Reduction Amount under Different Scenarios

First, determine the pollutant assessment target according to the location of the Bahao Bridge transboundary section, refer to the national surface water Class I–V corresponding to each pollutant concentration standard limit (Table 2). According to Equation (2), there are four kinds of pollutants exceeding the standard in this basin (Figure 3): permanganate index, total phosphorus, ammonia nitrogen, and fluoride. Combined with the "Functional Zones of Hebei Province" [25] jointly issued by the Hebei Provincial Department of Ecology and Environment, the administrative unit in which the watershed is located. This section is located on the inlet of Guanting Reservoir, which belongs to the Zhangjiakou water environment function zone of the Yang River. The minimum water quality standard should achieve Class III in this water environment function zone. Therefore, in scenario a, the water quality standard is set to Class III; in scenario b, the water quality standard is upgraded to Class II based on governance effectiveness and policies.

**Table 2.** Environmental quality standards for surface water in China (mg/L).

| Types of Pollutants | I | II | III | IV | V |
|:---:|:---:|:---:|:---:|:---:|:---:|
| COD$\leq$ | 15 | 15 | 20 | 20 | 30 |
| NH$_3$-H$\leq$ | 0.15 | 0.5 | 1.0 | 1.5 | 2.0 |
| BOD$_5\leq$ | 3 | 3 | 4 | 6 | 10 |
| TP$\leq$ | 0.02 | 0.1 | 0.2 | 0.3 | 0.4 |
| F$^-\leq$ | 1.0 | 1.0 | 1.0 | 1.5 | 2.0 |
| DO$\leq$ | 7.5 | 6 | 5 | 3 | 2 |
| pH$\leq$ | | | 6–9 | | |
| C$_u\leq$ | 0.01 | 1.0 | 1.0 | 1.0 | 1.0 |

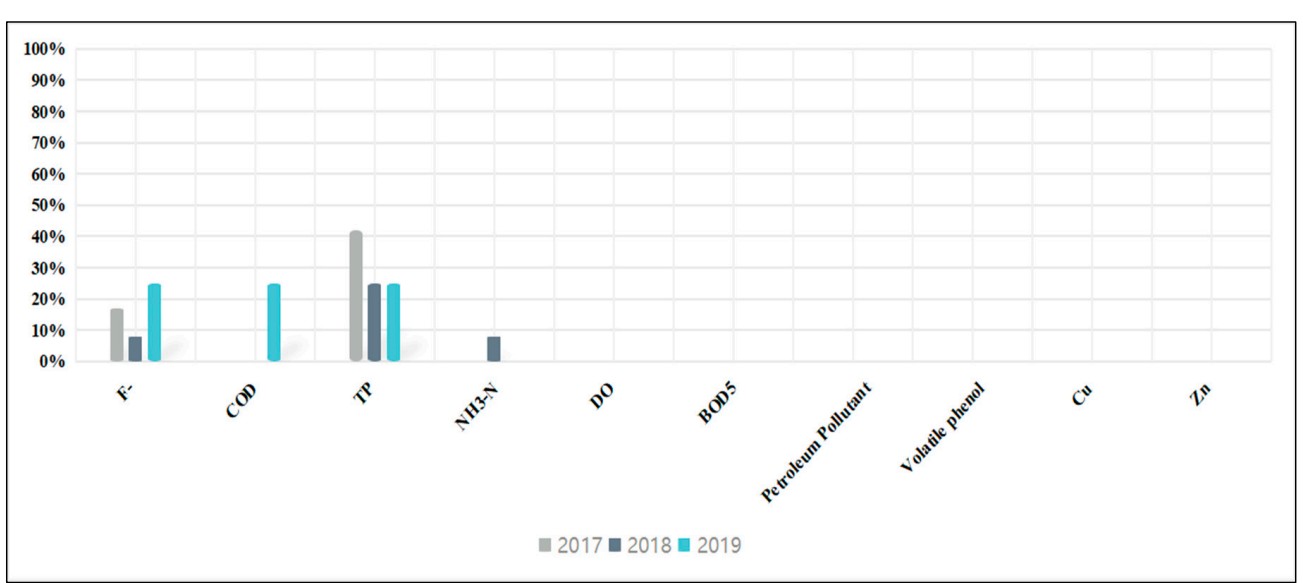

**Figure 3.** Excessive rate of pollutants in a section of the Bahao Bridge, 2017–2019.

### 3.1.1. Scenario Constructed with Actual Monitoring Data in 2018

Based on the addition of the monthly flow data of the Bahao Bridge section in 2018 the actual transit water yield in this year is 212 million m$^3$. According to the water quality monitoring monthly data of the Bahao Bridge section in 2018, Equation (3) is used to calculate the reduction of four excessive pollutants in the watershed. Table 3 shows results of the excessive pollutants limit transit volume $P_{0i}$, actual transit volume $P_{ai}$ and pollutant reduction volume $P_i$.

**Table 3.** Transit volume and reduction of excess pollutants in the Bahao Bridge section under the actual situation in 2018.

| Types of Pollutants | COD | TP | NH$_3$-N | F$^-$ |
|---|---|---|---|---|
| $P_{0i}$(t) | 5093 | 51 | 255 | 255 |
| $P_{ai}$(t) | 1178 | 37 | 101 | 173 |
| $P_i$(t) | 3915 | 14 | 154 | 82 |

### 3.1.2. Scenarios Constructed with Local Policies and Environmental Goals

Assuming that the concentration of pollutants in the Bahao Bridge section is raised to the level of Class II water, the reduction of the four types of pollutants exceeding the standard is calculated according to the predicted amount of transit water under different guarantee rates in the "Yongding river comprehensive treatment and the overall concept of ecological restoration" (Table 4).

**Table 4.** Transboundary volume and reduction of pollutants exceeding the standard in the Bahao Bridge section under the target scenario.

| Guarantee Rate of Outbound Water (%) | Outbound Water Yield (Million m$^3$) | Contaminants (t) | Types of Pollutants | | | |
|---|---|---|---|---|---|---|
| | | | COD | TP | NH$_3$-N | F$^-$ |
| 95% (Special dry year) | 90 | $P_{0i}$(t) | 1800 | 18 | 90 | 90 |
| | | $P_{ai}$(t) | 1350 | 9 | 45 | 90 |
| | | $P_i$(t) | 450 | 9 | 45 | 0 |
| 75% (General dry year) | 145 | $P_{0i}$(t) | 2900 | 29 | 145 | 145 |
| | | $P_{ai}$(t) | 2175 | 14.5 | 72.5 | 145 |
| | | $P_i$(t) | 725 | 14.5 | 72.5 | 0 |
| 50% (Normal year) | 209 | $P_{0i}$(t) | 4180 | 41.8 | 209 | 209 |
| | | $P_{ai}$(t) | 3135 | 20.9 | 104.5 | 209 |
| | | $P_i$(t) | 1045 | 20.9 | 104.5 | 0 |

The study area is the Hebei–Beijing section of the Yongding River. Therefore, following the downstream environmental protection tax standard in Beijing [26], the taxable pollutant tax is 2.2 dollar/equivalent. Combining Equation (5), according to the equivalent value of the four excess pollutants (Table 1) and the number of pollutants payable under the Beijing Environmental Protection Tax, we can calculate the treatment cost of excess pollutant reduction under the two scenarios, which is the water quality compensation amount (Table 5).

**Table 5.** The amount of ecological compensation based on the amount of pollutant reduction in different scenarios.

| | Scenario a | Scenario b | | |
|---|---|---|---|---|
| | | 95% (Special Dry Year) | 75% (Normal Dry Year) | 50% (Normal Year) |
| **Water Quality Compensation W$_P$ (USD million)** | 8.9 | 10.6 | 17.2 | 25.0 |

### 3.2. Water Ecological Compensation Amount Accounting Based on Restoration Cost Method under Different Scenarios

According to the diachronic curve analysis of the runoff of the Bahao Bridge in the Overall Plan, it can be seen that in a special dry year (guarantee rate of 95%), the outbound water volume of Q$_{95}$ is 90 million m$^3$, so it is set as the threshold for water volume compensation. The minimum limit for the amount of water left upstream must be reached at the threshold, only then can obtain the water quantity compensation fund. Analyzing the structure of agricultural, industrial, and domestic water consumption (Figure 4a) and

the contribution rate of water-saving (Figure 4b) in the upper reaches of Zhangjiakou City, we can see that agricultural water accounts for the largest proportion and has the highest water-saving potential.

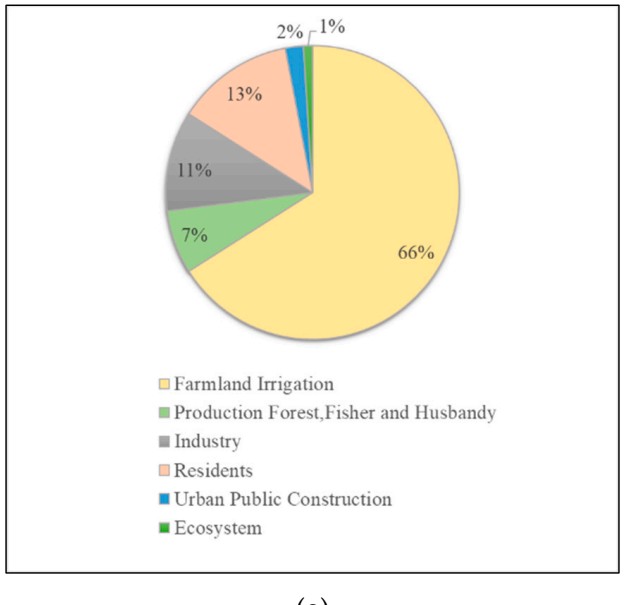

(**a**)

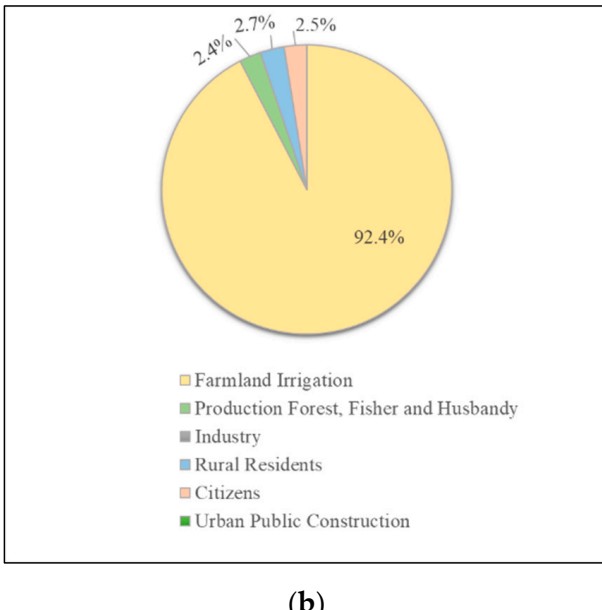

(**b**)

**Figure 4.** Water consumption structure (**a**) and proportion of water-saving potential (**b**) in Zhangjiakou.

Therefore, to achieve efficient water-saving, it is necessary to construct agricultural-related water-saving irrigation projects. Refering to the "Overall Plan", "Code for rational life and durability design of water resources and hydropower project, SL654-2014" [27], and "Yongding River Guarantee Agreement", the cost G of the planned water-saving irrigation project in the upper reaches of Zhangjiakou City and the additional water volume $V_m$ after the implementation of the projects can be estimated. According to Equations (9) and (10), it is calculated that the water quantity ecological compensation standard $C_y$ is 0.12 dollar/m³ (Table 6).

**Table 6.** Calculation of relevant parameters of water compensation standard based on restoration cost method.

| Water-Saving Irrigation Facilities (million dollar) | Annual Additional Water-Saving Capacity (10,000 m³) | Operation Years (Year) | Operation/Maintenance Cost (dollar/m³) | Water Compensation Standard (dollar/m³) |
|---|---|---|---|---|
| 132,823.5 | 3888 | ≥30 | 0.0062 | 0.12 |

Given that the threshold of the ecological compensation for water quantity is 90 million m³, the amount of transit water yield involved in the ecological compensation can be calculated under these two scenarios. According to local water transfer policie [28] and Equations (8), the water collection rate in the downstream area was about 60%. Finally, the amount of water ecological compensation in scenarios a and b (Table 7) can be obtained from Equations (7).

**Table 7.** The amount of water quantity ecological compensation based on restoration cost method under different scenarios.

| Scenarios | Scenario Description | The Amount of Water Quantity Ecological Compensation (USD Million) |
|---|---|---|
| a | Based on the measured data of the Bahao Bridge Section in 2018 | 24.2 |
| b | 95% (special dry year) | 0 |
| | 75% (General dry year) | 10.9 |
| | 50% (Normal year) | 23.4 |

*3.3. Accounting of Ecological Compensation Amount for Water Quality and Quantity in Different Scenarios*

For the Hebei–Beijing section of the Yongding River, the downstream demand for water resources is greater than the demand for water quality improvement. Therefore, considering the two aspects of water quality and water quantity, the coefficient that reflects the different compensation weights of water quality and quantity should be added to play a regulatory role. According to the "Zhangjiakou Statistical Bulletin of National Economic and Social Development in 2018" published by the Zhangjiakou City Statistics Bureau [29], the per capita water resources of Zhangjiakou City in 2018 were about 406 m$^3$. According to the "Beijing Statistical Bulletin on National Economic and Social Development in 2018" [30], the per capita water resources in Beijing in 2018 was about 165 m$^3$. We use Equation (11) to calculate the water compensation adjustment coefficient K as 2.46. According to the calculation of the water quality compensation amount and the water quantity compensation amount for scenarios a and b in Sections 3.1 and 3.2 above, combined with the water quality-water quantity compensation amount adjustment coefficient K, use Equation (1) to calculate the amount of ecological compensation of water quality and quantity (Table 8).

**Table 8.** The amount of ecological compensation of water quality and quantity in two scenarios.

| | Scenario a | Scenario b | | |
|---|---|---|---|---|
| | | 95% (Special Dry Year) | 75% (General Dry Year) | 50% (Normal Year) |
| The amount of transboundary ecological compensation (USD million) | 68.2 | 10.6 | 44.0 | 82.6 |

**4. Discussion**

*4.1. Relationship between Pollutant Concentration and the Change of Water Quantity*

The quality of the water environment is closely related to the change of water quantity in the watershed. For the rivers in northern China, the difference in rainfall distribution in the whole year is much higher than that in the southern region, so the water resource supply of rivers in flood season and non-flood season varies significantly in a year. Such changes in water quantity are usually reflected in changes in runoff. Although the water quantity in a watershed varies throughout the year, pollutant emissions are relatively constant throughout the year for the region. Due to the significant change in water quantity, the concentration of pollutants measured at each monitoring section has a great difference.

Gini coefficient is a model applied in the field of economics to describe whether the income distribution of people in a country/region is balanced. Its value is between 0 and 1, and the smaller the coefficient is, the more average the income distribution of people in the region is. On the contrary, the greater the difference of income distribution in the

region. In this study, Gini coefficient was applied to analyze the internal linkage between pollutant discharge and water quantity in the watershed: is used to describe the average distribution of pollutant quantity in the water environment of the watershed, that is, the fluctuation of pollutant concentration in the watershed during this period. The smaller the water quality–water quantity Gini coefficient is, the more curved the Lorentz curve is, indicating the more stable the water environmental quality state of the watershed is. Instead, the greater the fluctuation of the pollutant concentration in the watershed is and the water quality changes with the variation of water quantity. The stability of the received water quality is particularly important for the region in which the basin is located. Different water quality categories determine the different uses of water. The changing state of the water environment quality will affect the utilization planning of downstream water, and then affect the economic development and the water security of residents.

In this paper, the water quality–water quantity Gini coefficient of the Bahao Bridge section (115.4° E, 40.4° N) and Huaibin section (115.4° E, 32.4° N), Ji'an section (115.0° E, 27.0° N), Xiheyi section (115.4° E, 30.3° N) which are located in three different rivers from north to south are listed (Figure 5). The water quality–water quantity Gini coefficient in the southern regions with abundant water quantity is less than that in the northern regions with large seasonal variation of water quantity, indicating that compared with the southern region, the influence of water quantity on the pollutant concentration in the northern rivers is more severe.

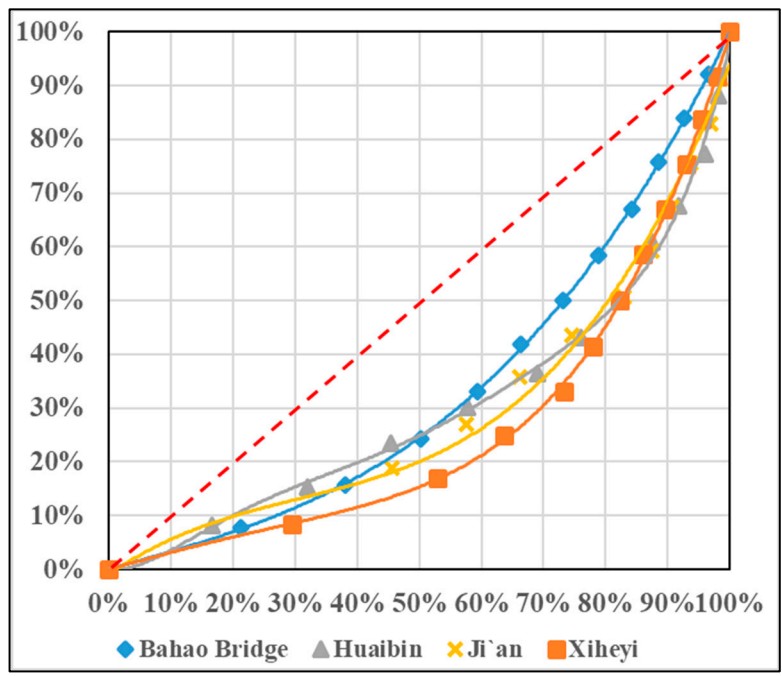

**Figure 5.** Comparison of the water quality–water quantity Gini coefficients.

We have also referred to the international research results on the relationship between pollutants and runoff in other transboundary river, such as the Neris (Viliya) River located on the border of Lithuania and Belarus, which is highly susceptible to changes in water quality and variations in the potential pollution load that could influence its eco-systems significantly. In the study of Marina et al. [31], the evaluation considered a decrease in river discharge due to changes in the regional storm-water flow. The decrease of runoff will lead to the increase of pollutant concentration in the river channel. They also pointed out that untreated storm-water flows can lead to increased concentrations of total phosphorus and nitrate nitrogen. Therefore, for areas with uneven distribution of rainfall or random occurrence of rainstorm events throughout the year, this is another factor affecting the stability of water quality and quantity. For northern areas, the construction of a storm-water

treatment plant can be considered at the same time as ecological water transfer, to reduce the negative impact of rainfall on water quality.

### 4.2. Advantages of the Ecological Compensation Standard of Water Quality and Quantity

Due to the significant difference of runoff in flood season and non-flood season, the annual mean of pollutant concentration cannot effectively reflect the flux and total amount of pollutants through the section. Compared with the accounting method of water quality compensation, which only considers annual pollutant concentration, the accounting method of quality joint compensation sets the water compensation adjustment coefficient according to the specific water resource endowment and difference of each stakeholder in the watershed, which reflects the demand of different regions for water resources in the watershed, avoids the "one size fits all" compensation method and expands the applicable scope of the accounting method.

Compared with the accounting results of water quality compensation, the accounting method of quality joint compensation has a larger scale of capital compensation. The already implemented Chaobai River Watershed Ecological compensation Project located the upstream of Miyun Reservoir in Beijing and the Anhui-Zhejiang Xin'a River Watershed Ecological compensation Project, which is also a case of provincial administrative transboundary watershed ecological compensation, both determined the scale of watershed ecological compensation funds to be hundreds of millions of dollar. Although the amount of water in the Yongding River watershed is less than that in the above cases, the Guanting Reservoir downstream is an important reserve water source of Beijing, so it also has important ecological and social-economic value. In contrast, the calculation results of quality joint compensation are closer to the scale of ecological compensation funds of various watersheds, and the method of quality joint compensation is more operable.

In scenario b, assuming that the water quality is II class, the outflow of water in normal years (50%) is about 40% higher than that in ordinary dry years (75% guarantee rate), with an absolute increase of 64 million m$^3$. Such a scale of water resources is very important for the Beijing–Tianjin–Hebei region where Yongding River watershed is located, which is equivalent to the actual amount of water discharged into the reservoir for a "Yellow River diversion". With a 40% increase in water quantity, the amount of water quality compensation increased by about 36%, while the amount of quality joint compensation increased by 110%. The substantial increase in compensation amount can better stimulate the provision of ecological compensation services for the compensation objects.

### 4.3. Disadvantages of the Ecological Compensation Standard of Water Quality and Quantity

The water quality and water quantity of the river in the watershed are affected by multiple factors such as climate change and human activities, and the monitoring results of water quality and water quantity in the assessment section have the characteristic of "multiple causes and one effect". The double compensation for water quality and water quantity should not be limited to the transboundary section. It can be comprehensively assessed according to the implementation of ecological compensation planning and the effectiveness of ecological compensation, and the focus should be shifted from upstream and downstream game to upstream and downstream cooperation.

According to the Opinions on Improving the Compensation Mechanism for Ecological Protection, by 2020, ecological protection compensation will be fully covered in key areas, areas prohibited from development, key ecological function zones, and other important areas. For a river watershed, it may involve several key areas and important regions, and it needs to connect the terrestrial ecosystem such as forest and the aquatic ecosystem such as a wetland. At present, the quality joint compensation is carried out according to the assessment section within the watershed, but the focus is still on the river and water area, and the separation between land and water has not been broken. Considering that other forms of ecological compensation, such as "Beijing–Tianjin–Hebei Water Source Project", "Beijing–Tianjin–Hebei Sand Source Project", or other similar ecological compensation

mechanisms have been implemented in the Yongding River watershed, there are still some repeated compensations in the Yongding River watershed, which is not conducive to improving the effectiveness of ecological compensation in the watershed.

At present, a variety of guidelines, implementation plans, and compensation methods have been introduced for watershed ecological compensation, and many regions have signed the upstream and downstream ecological compensation agreements or even the three parties. However, the watershed ecological compensation mechanism, which is mainly based on the "game" of transboundary water quality, needs to be innovated.

## 5. Conclusions

The single accounting method to investigate the water quality of the river watershed is not suitable for the ecological compensation of water-deficient rivers in the north. In this study, the ecological compensation method based on water quality and water quantity was used to carry out the accounting of watershed ecological compensation, which expands the fund scale, improves the comprehensiveness of watershed ecological compensation, and further meets the needs of all stakeholders of watershed ecological compensation. Integrating water quantity into the assessment scope of ecological compensation for transboundary watersheds will help further realize the effect of "Areas with large amounts of water output more water, while areas with little water output less water" and "Areas with good water quality export more water, while areas with poor water quality export less water". Compared with the current ecological water replenishment and water transfer systems in the Yongding River watershed, the scale of compensation funds and the scope of assessment have been significantly expanded.

According to the measured data of the Bahao Bridge section in 2018 (scenario a) and the target scenario (scenario b) constructed by the "Overall Plan", the amount of ecological compensation for water quality is USD 8.9 million, and the amount of water quality and quantity compensation is USD 68.2 million in scenario a. The amount of water quality compensation ranges from USD 10.6 million to USD 25.0 million, and the amount of water quality and quantity compensation ranges from USD 68 million to USD 82.6 million in Scenario b. It shows that the change of water quantity is more dominant in the expansion of the ecological compensation fund scale, while the change of water quality causes a small increase in the fund, which also reflects that the weight proportion of the water quantity in the compensation is higher than that of the water quality compensation.

The superiority of implementing ecological compensation for the water quality and quantity according to local conditions can be seen. However, from the perspective of the long-term economic and social development of the river basin, ecological compensation limited to cross-sections still cannot solve the problem of lagging effects of ecological environmental protection construction. This problem can be better solved by moving from single-section ecological compensation to comprehensive ecological compensation in the basin.

**Author Contributions:** Conceptualization, R.Y. and Y.W.; methodology, R.Y.; software, L.Z.; validation, X.L., Y.Z.; formal analysis, W.L.; resources, X.L.; data curation, Q.L.; writing—original draft preparation, Y.W.; writing—review and editing, X.L.; visualization, Y.L.; supervision, R.Y.; project administration, X.L.; funding acquisition, X.L. All authors have read and agreed to the published version of the manuscript.

**Funding:** This research was funded by [Major Science and Technology Program for Water Pollution Control and Treatment, China] grant number [2018ZX07111002]. And the APC was funded by [Major Science and Technology Program for Water Pollution Control and Treatment, China].

**Institutional Review Board Statement:** Not applicable.

**Informed Consent Statement:** Not applicable.

**Data Availability Statement:** The dataset in this study cannot be shared due to government and regulatory confidentiality in China.

**Acknowledgments:** This work was supported by the Major Science and Technology Program for Water Pollution Control and Treatment, China [grant numbers 2018ZX07111002]. The project "Study on ecological compensation of transboundary watershed in Hongjiannao nature reserve" (Beijing Normal University, China) is also acknowledged.

**Conflicts of Interest:** The authors declare no conflict of interest.

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
