# Peer review of "Study on Trans-Boundary Water Quality and Quantity Ecological Compensation Standard: A Case of the Bahao Bridge Section in Yongding River, China"

_water, doi:10.3390/w13111488_

Round 1

Reviewer 1 Report

In this work, Yizhuo et al. proposes a boundary water quality and quantity ecological compensation standard model.

Line 141: Study area- The GIS coordinates are lacking and so is the software used to make the map, this is something that requires attention

Line 175: are the equations symbols the ones that should be used? please redefine

Figure 2,4: The image is of poor quality

Line 196,206,231: yet again, the symbols should be changed

Line 453-464: Please limit the use of "Gini" marker, it was used intensively it makes the paper hard to read and unprofessional

Indeed water quality indicators cannot effectively compensate the ecosystem service providers for their expenditure on the environment and the paper adds value to the current literature.

Thus being said, the above mentioned issues and especially a general reformulation of the main body text is highly recommended

Reviewer 2 Report

Dear Authors, please indicate your novelty for this article in details. According to the scientific review from China you can compare yours research with best practice examples from European Rivers (e.g. Evaluating the Impacts of Integrated Pollution on Water Quality of the Trans-Boundary River) and/or similar articles.  Scenario constructed with actual monitoring data in 2017 must be extended till at least 2020 with newest measurements. 

Sincerely, Reviewer. 

Round 2

Reviewer 1 Report

The authors have addressed my concern and I now find this article, from my point of view, suitable for publication.